# Self-supervised GAN: Analysis and Improvement with Multi-class Minimax Game

**Ngoc-Trung Tran, Viet-Hung Tran, Ngoc-Bao Nguyen, Linxiao Yang, Ngai-Man Cheung**
Singapore University of Technology and Design (SUTD)

Corresponding author: Ngai-Man Cheung <`ngaiman_cheung@sutd.edu.sg`>

## Abstract

Self-supervised (SS) learning is a powerful approach for representation learning using unlabeled data. Recently, it has been applied to Generative Adversarial Networks (GAN) training. Specifically, SS tasks were proposed to address the catastrophic forgetting issue in the GAN discriminator. In this work, we perform an in-depth analysis to understand how SS tasks interact with learning of generator. From the analysis, we identify issues of SS tasks which allow a severely mode-collapsed generator to excel the SS tasks. To address the issues, we propose new SS tasks based on a multi-class minimax game. The competition between our proposed SS tasks in the game encourages the generator to learn the data distribution and generate diverse samples. We provide both theoretical and empirical analysis to support that our proposed SS tasks have better convergence property. We conduct experiments to incorporate our proposed SS tasks into two different GAN baseline models. Our approach establishes state-of-the-art FID scores on CIFAR-10, CIFAR-100, STL-10, CelebA, Imagenet $32 \times 32$ and Stacked-MNIST datasets, outperforming existing works by considerable margins in some cases. Our unconditional GAN model approaches performance of conditional GAN *without* using labeled data. Our code: `https://github.com/tntrung/msgan`

## 1 Introduction

**Generative Adversarial Networks (GAN).** GAN [12] have become one of the most important methods to learn generative models. GAN has shown remarkable results in various tasks, such as: image generation [17, 2, 18], image transformation [16, 53], super-resolution [23], text to image [38, 50], anomaly detection [41, 26]. The idea behind GAN is the mini-max game. It uses a binary classifier, so-called the discriminator, to distinguish the data (real) versus generated (fake) samples. The generator of GAN is trained to confuse the discriminator to classify the generated samples as the real ones. By having the generator and discriminator competing with each other in this adversarial process, they are able to improve themselves. The end goal is to have the generator capturing the data distribution. Although considerable improvement has been made for GAN under the conditional settings [34, 49, 2], i.e., using ground-truth labels to support the learning, it is still very challenging with unconditional setup. Fundamentally, using only a single signal (real/fake) to guide the generator to learn the high-dimensional, complex data distribution is very challenging [11, 1, 3, 5, 30, 40].

**Self-supervised Learning.** Self-supervised learning is an active research area [6, 35, 51, 52, 33, 10]. Self-supervised learning is a paradigm of unsupervised learning. Self-supervised methods encourage the classifier to learn better feature representation with *pseudo-labels*. In particular, these methods propose to learn image feature by training the model to recognize some geometric transformation that is applied to the image which the model receives as the input. A simple-yet-powerful method proposed in [10] is to use image rotations by 0, 90, 180, 270 degrees as the geometric transformation. The model is trained with the 4-way classification task of recognizing one of the four rotations. This

task is referred as the *self-supervised task*. This simple method is able to close the gap between supervised and unsupervised image classification [10].

**Self-supervised Learning for GAN.** Recently, self-supervised learning has been applied to GAN training [4, 44]. These works propose auxiliary self-supervised classification tasks to assist the main GAN task (Figure 1). In particular, their objective functions for learning discriminator $D$ and generator $G$ are multi-task loss as shown in (1) and (2) respectively:

$$\max_{D,C} \mathcal{V}_{\mathcal{D}}(D, C, G) = \mathcal{V}(D, G) + \lambda_d \Psi(G, C) \tag{1}$$

$$\min_{G} \mathcal{V}_{\mathcal{G}}(D, C, G) = \mathcal{V}(D, G) - \lambda_g \Phi(G, C) \tag{2}$$

$$\mathcal{V}(D, G) = \mathbb{E}_{\mathbf{x} \sim P_d} \log \Big( D(\mathbf{x}) \Big) + \mathbb{E}_{\mathbf{x} \sim P_g} \log \Big( 1 - D(\mathbf{x}) \Big) \tag{3}$$

Here, $\mathcal{V}(D, G)$ in (3) is the *GAN task*, which is the original value function proposed in Goodfellow et al. [12]. $P_d$ is true data distribution, $P_g$ is the distribution induced by the generator mapping. $\Psi(G, C)$ and $\Phi(G, C)$ are the *self-supervised (SS) tasks* for discriminator and generator learning, respectively (details to be discussed). $C$ is the classifier for the self-supervised task, e.g. rotation classifier as discussed [10]. Based on this framework, Chen et al.[4] apply self-supervised task to help discriminator counter catastrophic forgetting. Empirically, they have shown that self-supervised task enables discriminator to learn more stable and improved representation. Tran et al. [44] propose to improve self-supervised learning with adversarial training.

Despite the encouraging empirical results, in-depth analysis of the interaction between SS tasks ($\Psi(.)$ and $\Phi(.)$) and GAN task ($\mathcal{V}(D, G)$) has not been done before. On one hand, the application of SS task for *discriminator learning* is reasonable: the goal of discriminator is to classify real/fake image; an additional SS classification task $\Psi(G, C)$ could assist feature learning and enhance the GAN task. On the other hand, the motivation and design of SS task for *generator learning* is rather subtle: the goal of generator learning is to capture the data distribution in $G$, and it is unclear exactly how an additional SS classification task $\Phi(G, C)$ could help.

**In this work,** we conduct in-depth empirical and theoretical analysis to understand the interaction between self-supervised tasks ($\Psi(.)$ and $\Phi(.)$) and learning of generator $G$. Interestingly, from our analysis, we reveal issues of existing works. Specifically, the SS tasks of existing works have "loophole" that, during generator learning, $G$ could exploit to maximize $\Phi(G, C)$ without truly learning the data distribution. We show that analytically and empirically that a severely mode-collapsed generator can excel $\Phi(G, C)$. To address this issue, we propose new SS tasks based on a multi-class minimax game. Our proposed new SS tasks of discriminator and generator compete with each other to reach the equilibrium point. Through this competition, our proposed SS tasks are able to support the GAN task better. Specifically, our analysis shows that *our proposed SS tasks enhance matching between $P_d$ and $P_g$ by leveraging the transformed samples used in the SS classification* (rotated images when [10] is applied). In addition, our design couples GAN task and SS task. To validate our design, we provide theoretical analysis on the convergence property of our proposed SS tasks. Training a GAN with our proposed self-supervised tasks based on multi-class minimax game significantly improves baseline models. Overall, our system establishes state-of-the-art Fréchet Inception Distance (FID) scores. In summary, our contributions are:

- We conduct in-depth empirical and theoretical analysis to understand the issues of self-supervised tasks in existing works.
- Based on the analysis, we propose new self-supervised tasks based on a multi-class minimax game.
- We conduct extensive experiments to validate our proposed self-supervised tasks.

## 2  Related works

While training GAN with conditional signals (e.g., ground-truth labels of classes) has made good progress [34, 49, 2], training GAN in the unconditional setting is still very challenging. In the original GAN [12], the single signal (real or fake) of samples is provided to train discriminator and the generator. With these signals, the generator or discriminator may fall into ill-pose settings, and they

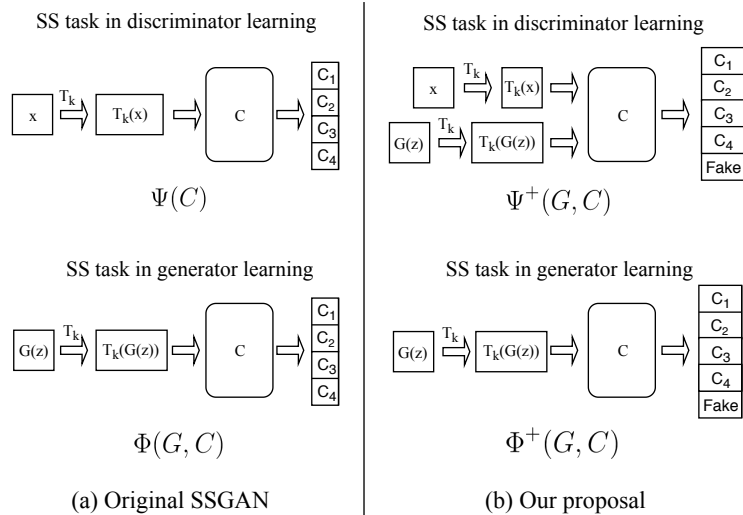

Figure 1: The model of (a) SSGAN [4] and (b) our approach. Here, $\Psi(C)$ and $\Phi(G, C)$ are the self-supervised value functions in training discriminator and generator, respectively, as proposed in [4]. $\Psi^+(G, C)$ and $\Phi^+(G, C)$ are the self-supervised value functions proposed in this work.

may get stuck at bad local minimums though still satisfying the signal constraints. To overcome the problems, many regularizations have been proposed. One of the most popular approaches is to enforce (towards) Lipschitz condition of the discriminator. These methods include weight-clipping [1], gradient penalty constraints [13, 39, 21, 36, 27] and spectral norm [31]. Constraining the discriminator mitigates gradients vanishing and avoids sharp decision boundary between the real and fake classes.

Using Lipschitz constraints improve the stability of GAN. However, the challenging optimization problem still remains when using a single supervisory signal, similar to the original GAN [12]. In particular, the learning of discriminator is highly dependent on generated samples. If the generator collapses to some particular modes of data distribution, it is only able to create samples around these modes. There is no competition to train the discriminator around other modes. As a result, the gradients of these modes may vanish, and it is impossible for the generator to model well the entire data distribution. Using additional supervisory signals helps the optimization process. For example, using self-supervised learning in the form of auto-encoder has been proposed. AAE [29] guides the generator towards resembling realistic samples. However, an issue with using auto-encoder is that pixel-wise reconstruction with $\ell_2$-norm causes blurry artifacts. VAE/GAN [22], which combining VAE [19] and GAN, is an improved solution: while the discriminator of GAN enables the usage of feature-wise reconstruction to overcome the blur, the VAE constrains the generator to mitigate mode collapse. In ALI [8] and BiGAN [7], they jointly train the data/latent samples in the GAN framework. InfoGAN [5] infers the disentangled latent representation by maximizing the mutual information. In [42, 43], they combine two different types of supervisory signals: real/fake signals and self-supervised signal in the form of auto-encoder. In addition, Auto-encoder based methods, including [22, 42, 43], can be considered as an approach to mitigate catastrophic forgetting because they regularize the generator to resemble the real ones. It is similar to EWC [20] or IS [48] but the regularization is achieved via the output, not the parameter itself. Although using feature-wise distance in auto-encoder could reconstruct sharper images, it is still challenging to produce very realistic detail of textures or shapes.

Several different types of supervisory signal have been proposed. Instead of using only one discriminator or generator, they propose ensemble models, such as multiple discriminators [32], mixture of generators [15, 9] or applying an attacker as a new player for GAN training [28]. Recently, training model with auxiliary self-supervised constraints [4, 44] via multi pseudo-classes [10] helps improve stability of the optimization process. This approach is appealing: it is simple to implement and does not require more parameters in the networks (except a small head for the classifier). Recent work applies InfoMax principle to improve GAN [24]. Variational Autoencoder is another important approach to learn generative models [19, 46].

# 3 GAN with Auxiliary Self-Supervised tasks

In [4], self-supervised (SS) value function (also referred as "self-supervised task") was proposed for GAN [12] via image rotation prediction [10]. In their work, they showed that the SS task was useful to mitigate catastrophic forgetting problem of GAN discriminator. The objectives of the discriminator and generator in [4] are shown in Eq. 4 and 5. Essentially, the SS task of the discriminator (denoted by $\Psi(C)$) is to train the classifier $C$ that maximizes the performance of predicting the rotation applied to the *real* samples. Given this classifier $C$, the SS task of the generator (denoted by $\Phi(G, C)$) is to train the generator $G$ to produce *fake* samples for maximizing classification performance. The discriminator and classifier are the same (shared parameters), except the last layer in order to implement two different heads: the last fully-connected layer which returns a one-dimensional output (real or fake) for the discriminator, and the other which returns a $K$-dimensional softmax of pseudo-classes for the classifier. $\lambda_d$ and $\lambda_g$ are constants.

$$\max_{D,C} \mathcal{V}(D, C, G) = \mathcal{V}(D, G) + \lambda_d \underbrace{\left( \mathbb{E}_{\mathbf{x} \sim P_d^T} \mathbb{E}_{T_k \sim \mathcal{T}} \log \left( C_k(\mathbf{x}) \right) \right)}_{\Psi(C)} \tag{4}$$

$$\min_{G} \mathcal{V}(D, C, G) = \mathcal{V}(D, G) - \lambda_g \underbrace{\left( \mathbb{E}_{\mathbf{x} \sim P_g^T} \mathbb{E}_{T_k \sim \mathcal{T}} \log \left( C_k(\mathbf{x}) \right) \right)}_{\Phi(G,C)} \tag{5}$$

Here, the GAN value function $\mathcal{V}(D, G)$ (also referred as "GAN task") can be the original minimax GAN objective [12] or other improved versions. $\mathcal{T}$ is the set of transformation, $T_k \in \mathcal{T}$ is the $k$-th transformation. The rotation SS task proposed in [10] is applied, and $T_1, T_2, T_3, T_4$ are the 0, 90, 180, 270 degree image rotation, respectively. $P_d, P_g$ are the distributions of real and fake data samples, respectively. $P_d^T, P_g^T$ are the mixture distribution of *rotated* real and fake data samples (by $T_k \in \mathcal{T}$), respectively. Let $C_k(\mathbf{x})$ be the $k$-th softmax output of classifier $C$, and we have $\sum_{k=1}^{K} C_k(\mathbf{x}) = 1, \forall \mathbf{x}$. The models are shown in Fig. 1a. In [4], empirical evidence of improvements has been provided.

Note that, the goal of $\Phi(G, C)$ is to encourage the generator to produce realistic images. It is because classifier $C$ is trained with real images and captures features that allow detection of rotation. However, the interaction of $\Phi(G, C)$ with the GAN task $\mathcal{V}(D, G)$ has not been adequately analyzed.

# 4 Analysis on Auxiliary Self-supervised Tasks

We analyze the SS tasks in [4] (Figure 1a). We assume that all networks $D, G, C$ have enough capacity [12]. Refer to the Appendix A for full derivation. Let $D^*$ and $C^*$ be the optimal discriminator and optimal classifier respectively at an equilibrium point. We assume that we have an optimal $D^*$ of the GAN task. We focus on $C^*$ of SS task. Let $p^{T_k}(\mathbf{x})$ be the probability of sample $\mathbf{x}$ under transformation by $T_k$ (Figure 2). $p_d^{T_k}(\mathbf{x}), p_g^{T_k}(\mathbf{x})$ denotes the probability $p^{T_k}(\mathbf{x})$ of data sample $(\mathbf{x} \sim P_d^T)$ or generated sample $(\mathbf{x} \sim P_g^T)$ respectively.

**Proposition 1** *The optimal classifier $C^*$ of Eq. 4 is:*

$$C_k^*(\mathbf{x}) = \frac{p_d^{T_k}(\mathbf{x})}{\sum_{k=1}^{K} p_d^{T_k}(\mathbf{x})} \tag{6}$$

*Proof.* Refer to our proof in Appendix A for optimal $C^*$.

**Theorem 1** *Given optimal classifier $C^*$ for SS task $\Psi(C)$, at the equilibrium point, maximizing SS task $\Phi(G, C^*)$ of Eq. 5 is equal to maximizing:*

$$\Phi(G, C^*) = \frac{1}{K} \sum_{k=1}^{K} \left[ \mathbb{E}_{\mathbf{x} \sim P_g^{T_k}} \log \left( \frac{p_d^{T_k}(\mathbf{x})}{\sum_{k=1}^{K} p_d^{T_k}(\mathbf{x})} \right) \right] = \frac{1}{K} \sum_{k=1}^{K} \mathcal{V}_{\Phi}^{T_k}(\mathbf{x}) \tag{7}$$

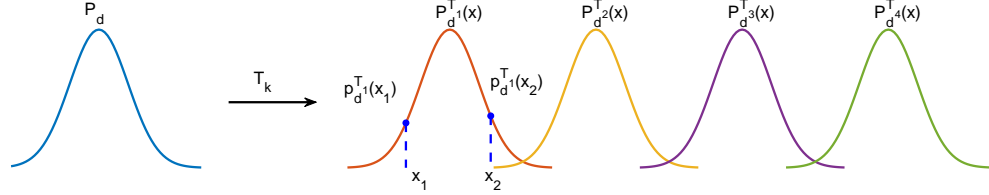

Figure 2: The probability distribution $p_d^{T_k}(\mathbf{x})$. Here, samples from $P_d$ are rotated by $T_k$. The distribution of rotated sample is $p^{T_k}(\mathbf{x})$. Some rotated samples resemble the original samples, e.g. those on the right of $\mathbf{x}_2$. On the other hand, for some image, there is no rotated image resembling it, e.g. $\mathbf{x}_1$ ($p_d^{T_j}(\mathbf{x}_1) = 0, j \neq 1$). The generator can learn to generate these images e.g. $\mathbf{x}_1$ to achieve maximum of $\Phi(G, C^*)$, without actually learning the entire $P_d$.

*Proof.* Refer to our proof in Appendix A.

Theorem 1 depicts learning of generator $G$ given the optimal $C^*$: selecting $G$ (hence $P_g$) to maximize $\Phi(G, C^*)$. As $C^*$ is trained on real data, $\Phi(G, C^*)$ encourages $G$ to learn to generate realistic samples. However, we argue that $G$ can maximize $\Phi(G, C^*)$ without actually learning data distribution $P_d$. *In particular, it is sufficient for $G$ to maximize $\Phi(G, C^*)$ by simply learning to produce images which rotated version is rare (near zero probability).* Some example images are shown in Figure 3a. Intuitively, for these images, rotation can be easily recognized.

The argument can be developed from Theorem 1. From (7), it can be shown that $\mathcal{V}_\Phi^{T_k}(\mathbf{x}) \leq 0$ ($p_g^{T_k}(\mathbf{x}) >= 0$ and $\frac{p_d^{T_k}(\mathbf{x})}{\sum_{k=1}^{K} p_d^{T_k}(\mathbf{x})} \leq 1$). One way for $G$ to achieve the maximum is to generate $\mathbf{x}$ such that $p_d^{T_1}(\mathbf{x}) \neq 0$ and $p_d^{T_j}(\mathbf{x}) = 0, j \neq 1$. For these $\mathbf{x}$, the maximum $\mathcal{V}_\Phi^{T_k}(\mathbf{x}) = 0$ is attained. Note that $T_1$ corresponds to 0 degree rotation, i.e., no rotation. Recall that $p_d^{T_k}(\mathbf{x})$ is the probability distribution of transformed data by $T_k$. Therefore the condition $p_d^{T_1}(\mathbf{x}) \neq 0$ and $p_d^{T_j}(\mathbf{x}) = 0, j \neq 1$ means that there is no other rotated image resembling $\mathbf{x}$, or equivalently, rotated $\mathbf{x}$ does not resemble any other images (Figure 2). Therefore, the generator can exploit this "loophole" to maximize $\Phi(G, C^*)$ without actually learning the data distribution. In particular, even a mode-collapsed generator can achieve the maximum of $\Phi(G, C^*)$ by generating such images.

**Empirical evidence.** Empirically, our experiments (in Appendix B.2.1) show that the FID of the models when using $\Phi(G, C)$ is poor except for very small $\lambda_g$. We further illustrate this issue by a toy empirical example using CIFAR-10. We augment the training images $\mathbf{x}$ with transformation data $T_k(\mathbf{x})$ to train the classifier $C$ to predict the rotation applied to $\mathbf{x}$. This is the SS task of discriminator in Figure 1a. Given this classifier $C$, we simulate the SS task of generator learning as follows. To simulate the output of a good generator $G_{good}$ which generates diverse realistic samples, we choose the full test set of CIFAR-10 (10 classes) images and compute the cross-entropy loss, i.e. $-\Phi(G, C)$, when they are fed into $C$. To simulate the output of a mode-collapsed generator $G_{collapsed}$, we select samples from *one* class, e.g. "horse", and compute the cross-entropy loss when they are fed into $C$. Fig. 3b show that some $G_{collapsed}$ can outperform $G_{good}$ and achieve a smaller $-\Phi(G, C)$. E.g. a $G_{collapsed}$ that produces *only* "horse" samples outperform $G_{good}$ under $\Phi(G, C)$. This example illustrates that, while $\Phi(G, C)$ may help the generator to create more realistic samples, it does not help the generator to prevent mode collapse. *In fact, as part of the multi-task loss (see (5)), $\Phi(G, C)$ would undermine the learning of synthesizing diverse samples in the GAN task $\mathcal{V}(D, G)$.*

# 5 Proposed method

## 5.1 Auxiliary Self-Supervised Tasks with Multi-class Minimax Game

In this section, we propose improved SS tasks to address the issue (Fig. 1b). Based on a multi-class minimax game, our classifier learns to distinguish the rotated samples from real data versus those from generated data. Our proposed SS tasks are $\Psi^+(G, C)$ and $\Phi^+(G, C)$ in (8) and (9) respectively.

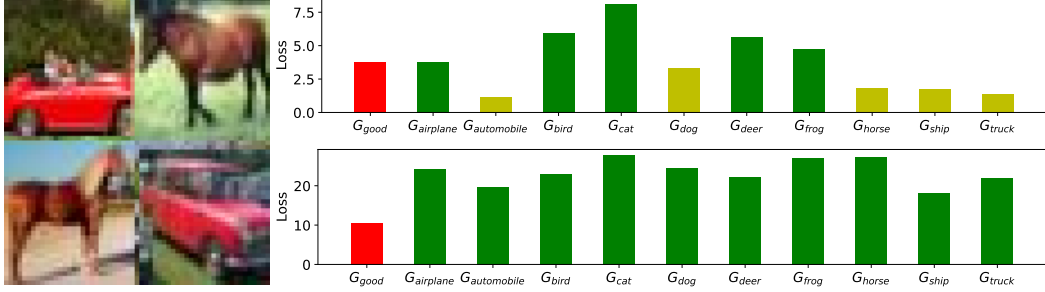

Figure 3: (a) Left: Example images that achieve minimal loss (or maximal $\Phi(G, C)$). For these images, rotation can be easily recognized: an image with a 90 degree rotated horse is likely due to applying $T_2$ rather than an original one. (b) Right (Top): the loss of original SS task, i.e. $-\Phi(G, C)$ computed over a good generator (red) and collapsed generators (green, yellow). Some collapsed generators (e.g. one that generates only "horse") have smaller loss than the good generator under $-\Phi(G, C)$. (c) Right (Bottom): the loss of proposed MS task, $-\Phi^+(G, C)$, of a good generator (red) and collapsed generators (green). The good generator has the smallest loss under $-\Phi^+(G, C)$.

Our discriminator objective is:

$$\max_{D,C} \mathcal{V}(D, C, G) = \mathcal{V}(D, G) + \lambda_d \underbrace{\left( \mathbb{E}_{\mathbf{x} \sim P_d^T} \mathbb{E}_{T_k \sim \mathcal{T}} \log \left( C_k(\mathbf{x}) \right) + \mathbb{E}_{\mathbf{x} \sim P_g^T} \mathbb{E}_{T_k \sim \mathcal{T}} \log \left( C_{K+1}(\mathbf{x}) \right) \right)}_{\Psi^+(G,C)}$$

(8)

Eq. 8 means that we simultaneously distinguish generated samples, as the $(K + 1)$-th class, from the rotated real sample classes. Here, $C_{K+1}(\mathbf{x})$ is the $(K + 1)$-th output for the fake class of classifier $C$.

While rotated real samples are fixed samples that help prevent the classifier (discriminator) from forgetting, the class $K + 1$ serves as the connecting point between generator and classifier, and the generator can directly challenge the classifier. Our technique resembles the original GAN by Goodfellow et al. [12], but we generalize it for multi-class minimax game. Our generator objective is:

$$\min_{G} \mathcal{V}(D, C, G) = \mathcal{V}(D, G) - \lambda_g \underbrace{\left( \mathbb{E}_{\mathbf{x} \sim P_g^T} \mathbb{E}_{T_k \sim \mathcal{T}} \log \left( C_k(\mathbf{x}) \right) - \mathbb{E}_{\mathbf{x} \sim P_g^T} \mathbb{E}_{T_k \sim \mathcal{T}} \log \left( C_{K+1}(\mathbf{x}) \right) \right)}_{\Phi^+(G,C)}$$

(9)

$\Psi^+(G, C)$ and $\Phi^+(G, C)$ form a multi-class minimax game. Note that, when we mention multi-class minimax game (or multi-class adversarial training), we refer to the SS tasks. The game for GAN task is the original by Goodfellow et al. [12].

### 5.1.1 Theoretical Analysis

**Proposition 2** *For fixed generator $G$, the optimal solution $C^*$ under Eq. 8 is:*

$$C_k^*(\mathbf{x}) = \frac{p_d^T(\mathbf{x})}{p_g^T(\mathbf{x})} \frac{p_d^{T_k}(\mathbf{x})}{\sum_{k=1}^{K} p_d^{T_k}(\mathbf{x})} C_{K+1}^*(\mathbf{x})$$

(10)

*where $p_d^T(\mathbf{x})$ and $p_g^T(\mathbf{x})$ are probability of sample $\mathbf{x}$ in the mixture distributions $P_d^T$ and $P_g^T$ respectively.*

*Proof.* Refer to our proof in Appendix A for optimal $C^*$.

**Theorem 2** *Given optimal classifier $C^*$ obtained from multi-class minimax training $\Psi^+(G, C)$, at the equilibrium point, maximizing $\Phi^+(G, C^*)$ is equal to maximizing Eq. 11:*

$$\Phi^+(G, C^*) = -\frac{1}{K} \left[ \sum_{k=1}^{K} \mathrm{KL}(P_g^{T_k} || P_d^{T_k}) \right] + \frac{1}{K} \sum_{k=1}^{K} \left[ \mathbb{E}_{\mathbf{x} \sim P_g^{T_k}} \log \left( \frac{p_d^{T_k}(\mathbf{x})}{\sum_{k=1}^{K} p_d^{T_k}(\mathbf{x})} \right) \right]$$

(11)

*Proof.* Refer to our proof in Appendix A.

Note that proposed SS task objective (11) is different from the original SS task objective (7) with the KL divergence term. Furthermore, note that $\text{KL}(P_g^{T_k}||P_d^{T_k}) = \text{KL}(P_g||P_d)$, as rotation $T_k$ is an affine transform and KL divergence is invariant under affine transform (our proof in Appendix A). Therefore, the improvement is clear: *Proposed SS tasks $\left(\Psi^+(.), \Phi^+(.)\right)$ work together to improve the matching of $P_g$ and $P_d$ by leveraging the rotated samples.* For a given $P_g$, feedbacks are computed from not only $\text{KL}(P_g||P_d)$ but also $\text{KL}(P_g^{T_k}||P_d^{T_k})$ via the rotated samples. Therefore, $G$ has more feedbacks to improve $P_g$. We investigate the improvement of our method on toy dataset as in Section 4. The setup is the same, except that now we replace models/cost functions of $-\Phi(G, C)$ with our proposed ones $-\Phi^+(G, C)$ (the design of $G_{good}$ and $G_{collapsed}$ are the same). The loss now is shown in Fig. 3c. Comparing Fig. 3c and Fig. 3b, the improvement using our proposed model can be observed: $G_{good}$ has the lowest loss under our proposed model. Note that, since optimizing KL divergence is not easy because it is asymmetric and could be biased to one direction [32], in our implementation, we use a slightly modified version as described in the Appendix.

## 6 Experiments

We measure the diversity and quality of generated samples via the Fréchet Inception Distance (FID) [14]. FID is computed with 10K real samples and 5K generated samples exactly as in [31] if not precisely mentioned. We report the best FID attained in 300K iterations as in [45, 25, 42, 47]. We integrate our proposed techniques into two baseline models (SSGAN [4] and Dist-GAN [42]). We conduct experiments mainly on CIFAR-10 and STL-10 (resized into $48 \times 48$ as in [31]). We also provide additional experiments of CIFAR-100, Imagenet $32 \times 32$ and Stacked-MNIST.

For Dist-GAN [42], we evaluate three versions implemented with different network architectures: DCGAN architecture [37], CNN architectures of SN-GAN [31] (referred as SN-GAN architecture) and ResNet architecture [13]. We recall these network architectures in Appendix C. We use ResNet architecture [13] for experiments of CIFAR-100, Imagenet $32 \times 32$, and tiny K/4, K/2 architectures [30] for Stacked MNIST. We keep all parameters suggested in the original work and focus to understand the contribution of our proposed techniques. For SSGAN [4], we use the ResNet architecture as implemented in the official code[1].

In our experiments, we use **SS** to denote the original self-supervised tasks proposed in [4], and we use **MS** to denote our proposed self-supervised tasks "Multi-class mini-max game based Self-supervised tasks". Details of the experimental setup and network parameters are discussed in Appendix B.

We have conducted extensive experiments. Setup and results are discussed in Appendix B. In this section, we highlight the main results:

- Comparison between **SS** and our proposed **MS** using the same baseline.
- Comparison between our proposed **baseline + MS** and other state-of-the-art unconditional and conditional GAN. We emphasize that our proposed **baseline + MS** is unconditional and does not use any label.

### 6.1 Comparison between SS and our proposed MS using the same baseline

Results are shown in Fig. 4 using Dist-GAN [42] as the baseline. For each experiment and for each approach (**SS** or **MS**), we obtain the best $\lambda_g$ and $\lambda_d$ using extensive search (see Appendix B.4 for details), and we use the best $\lambda_g$ and $\lambda_d$ in the comparison depicted in Fig. 4. In our experiments, we observe that Dist-GAN has stable convergence. Therefore, we use it in these experiments. As shown in Fig. 4, our proposed **MS** outperforms the original **SS** consistently. More details can be found in Appendix B.4.

### 6.2 Comparison between our proposed method with other state-of-the-art GAN

Main results are shown in Table 1. Details of this comparison can be found in Appendix B.4. The best $\lambda_g$ and $\lambda_d$ as in Figure 4 are used in this comparison. The best FID attained in 300K iterations

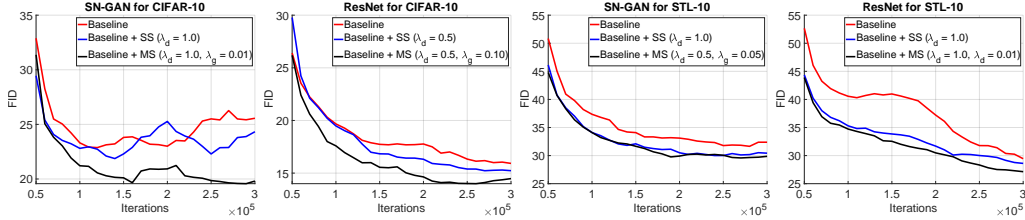

Figure 4: Compare **SS** (original SS tasks proposed in [4]) and **MS** (our proposed Multi-class mini-max game based Self-supervised tasks). The **baseline** is Dist-GAN [42], implemented with **SN-GAN** networks (CNN architectures in [31]) and **ResNet**. Two datasets are used, CIFAR-10 and STL-10. For each experiment, we use the best $\lambda_d, \lambda_g$ for the models, obtained through extensive search (Appendix B.4). Note that $\lambda_g = 0$ is the best for "Baseline + SS" in all experiments. The results suggest consistent improvement using our proposed self-supervised tasks.

Table 1: Comparison with other state-of-the-art GAN on CIFAR-10 and STL-10 datasets. We report the best FID of the methods. Two network architectures are used: **SN-GAN** networks (CNN architectures in [31]) and **ResNet**. The FID scores are extracted from the respective papers when available. **SS** denotes the original SS tasks proposed in [4]. **MS** denotes our proposed self-supervised tasks. '*': FID is computed with 10K-10K samples as in [4]. All compared GAN are unconditional, except SAGAN and BigGAN. SSGAN[+] is SS-GAN in [4] but using the best parameters we have obtained. In SSGAN[+] + MS, we replace the original **SS** in author's code with our proposed **MS**.

| Methods | SN-GAN | | ResNet | | |
| | CIFAR-10 | STL-10 | CIFAR-10 | STL-10 | CIFAR-10* |
| --- | --- | --- | --- | --- | --- |
| GAN-GP [31] | 37.7 | - | - | - | - |
| WGAN-GP [31] | 40.2 | 55.1 | - | - | - |
| SN-GAN [31] | 25.5 | 43.2 | $21.70 \pm .21$ | $40.10 \pm .50$ | 19.73 |
| SS-GAN [4] | - | - | - | - | 15.65 |
| Dist-GAN [42] | 22.95 | 36.19 | $17.61 \pm .30$ | $28.50 \pm .49$ | 13.01 |
| GN-GAN [43] | 21.70 | 30.80 | $16.47 \pm .28$ | - | - |
| SAGAN [49] (cond.) | - | - | 13.4 (best) | - | - |
| BigGAN [2] (cond.) | - | - | 14.73 | - | - |
| SSGAN[+] | - | - | - | - | 20.47 |
| **Ours(SSGAN[+] + MS)** | - | - | - | - | 19.89 |
| Dist-GAN + SS | 21.40 | 29.79 | $14.97 \pm .29$ | $27.98 \pm .38$ | 12.37 |
| **Ours(Dist-GAN + MS)** | **18.88** | **27.95** | $\mathbf{13.90 \pm .22}$ | $\mathbf{27.10 \pm .34}$ | **11.40** |

are reported as in [45, 25, 42, 47]. Note that SN-GAN method [31] attains the best FID at about 100K iterations with ResNet and it diverges afterward. Similar observation is also discussed in [4].

As shown in Table 1, our method (Dist-GAN + MS) consistently outperforms the baseline Dist-GAN and other state-of-the-art GAN. These results confirm the effectiveness of our proposed self-supervised tasks based on multi-class minimax game.

We have also extracted the FID reported in [4], i.e. SSGAN with the original SS tasks proposed there. In this case, we follow exactly their settings and compute FID using 10K real samples and 10K fake samples. Our model achieves better FID score than SSGAN with exactly the same ResNet architecture on CIFAR-10 dataset. See results under the column CIFAR-10* in Table 1.

Note that we have tried to reproduce the results of SSGAN using its published code, but we were unable to achieve similar results as reported in the original paper [4]. We have performed extensive search and we use the obtained best parameter to report the results as SSGAN[+] in Table 1 (i.e., SSGAN[+] uses the published code and the best parameters we obtained). We use this code and setup to compare **SS** and **MS**, i.e. we replace the **SS** code in the system with **MS** code, and obtain "SSGAN[+] + MS". As shown in Table 1, our "SSGAN[+] + MS" achieves better FID than SSGAN[+]. The improvement is consistent with Figure 4 when Dist-GAN is used as the baseline. More detailed experiments can be found in the Appendix. We have also compared SSGAN[+] and our system (SSGAN[+] + MS) on CelebA ($64 \times 64$). In this experiment, we use a small DCGAN architecture provided in the authors' code. Our proposed MS outperforms the original SS, with FID improved from 35.03 to 33.47. This experiment again confirms the effectiveness of our proposed MS.

Table 2: Results on CIFAR-100 and ImageNet 32×32. We use baseline model Dist-GAN with ResNet architecture. We follow the same experiment setup as above. **SS**: proposed in [4]; **MS**: this work.

| Datasets | SS | MS |
|---|---|---|
| CIFAR-100 (10K-5K FID) | 21.02 | 19.74 |
| ImageNet 32×32 (10K-10K FID) | 17.1 | 12.3 |

Table 3: Comparing to state-of-the-art methods on Stacked MNIST with tiny $K/4$ and $K/2$ architectures [30]. We also follow the same experiment setup of [30]. Baseline model: Dist-GAN. **SS**: proposed in [4]; **MS**: this work. Our method **MS** achieves the best results for this dataset with both architectures, outperforming state-of-the-art [42, 17] by a significant margin.

| Arch | Unrolled GAN [30] | WGAN-GP [13] | Dist-GAN [42] | Pro-GAN [17] | [42]+SS | Ours([42]+MS) |
|---|---|---|---|---|---|---|
| K/4, # | $372.2 \pm 20.7$ | $640.1 \pm 136.3$ | $859.5 \pm 68.7$ | $859.5 \pm 36.2$ | $906.75 \pm 26.15$ | $926.75 \pm 32.65$ |
| K/4, KL | $4.66 \pm 0.46$ | $1.97 \pm 0.70$ | $1.04 \pm 0.29$ | $1.05 \pm 0.09$ | $0.90 \pm 0.13$ | $0.78 \pm 0.13$ |
| K/2, # | $817.4 \pm 39.9$ | $772.4 \pm 146.5$ | $917.9 \pm 69.6$ | $919.8 \pm 35.1$ | $957.50 \pm 31.23$ | $976.00 \pm 10.04$ |
| K/2, KL | $1.43 \pm 0.12$ | $1.35 \pm 0.55$ | $1.06 \pm 0.23$ | $0.82 \pm 0.13$ | $0.61 \pm 0.15$ | $0.52 \pm 0.07$ |

We conduct additional experiments on CIFAR-100 and ImageNet 32×32 to compare **SS** and **MS** with Dist-GAN baseline. We use the same ResNet architecture as Section B.4 on CIFAR-10 for this study, and we use the best parameters $\lambda_d$ and $\lambda_g$ selected in Section B.4 for ResNet architecture. Experimental results in Table 2 show that our **MS** consistently outperform **SS** for all benchmark datasets. For ImageNet 32×32 we report the *best* FID for **SS** because the model suffers serious mode collapse at the end of training. Our **MS** achieves the best performance at the end of training.

We also evaluate the diversity of our generator on Stacked MNIST [30]. Each image of this dataset is synthesized by stacking any three random MNIST digits. We follow exactly the same experiment setup with tiny architectures $K/4$, $K/2$ and evaluation protocol of [30]. We measure the quality of methods by the number of covered modes (higher is better) and KL divergence (lower is better). Refer to [30] for more details. Table. 3 shows that our proposed **MS** outperforms **SS** for both mode number and KL divergence. Our approach significantly outperforms state-of-the-art [42, 17]. The means and standard deviations of **MS** and **SS** are computed from eight runs (we re-train our GAN model from the scratch for each run). The results are reported with best $(\lambda_d, \lambda_g)$ of **MS**: $(0.5, 0.2)$ for $K/4$ architecture and $(1.0, 1.0)$ for $K/2$ architecture. Similarly, best $(\lambda_d, \lambda_g)$ of **SS**: $(0.5, 0.0)$ for $K/4$ architecture and $(1.0, 0.0)$ for $K/2$ architecture.

Finally, in Table 1, we compare our FID to SAGAN [49] (a state-of-the-art conditional GAN) and BigGAN [2]. We perform the experiments under the same conditions using ResNet architecture on the CIFAR-10 dataset. We report the best FID that SAGAN can achieve. As SAGAN paper does not have CIFAR-10 results [49], we run the published SAGAN code and select the best parameters to obtain the results for CIFAR-10. For BigGAN, we extract best FID from original paper. Although our method is unconditional, our best FID is very close to that of these state-of-the-art conditional GAN. This validates the effectiveness of our design. Generated images using our system can be found in Figures 5 and 6 of Appendix B.

# 7 Conclusion

We provide theoretical and empirical analysis on auxiliary self-supervised task for GAN. Our analysis reveals the limitation of the existing work. To address the limitation, we propose multi-class minimax game based self-supervised tasks. Our proposed self-supervised tasks leverage the rotated samples to provide better feedback in matching the data and generator distributions. Our theoretical and empirical analysis support improved convergence of our design. Our proposed SS tasks can be easily incorporated into existing GAN models. Experiment results suggest that they help boost the performance of baseline implemented with various network architectures on the CIFAR-10, CIFAR-100, STL-10, CelebA, Imagenet $32 \times 32$, and Stacked-MNIST datasets. The best version of our proposed method establishes state-of-the-art FID scores on all these benchmark datasets.

## Acknowledgements

This work was supported by ST Electronics and the National Research Foundation(NRF), Prime Minister's Office, Singapore under Corporate Laboratory @ University Scheme (Programme Title: STEE Infosec - SUTD Corporate Laboratory). This research was also supported by the National Research Foundation Singapore under its AI Singapore Programme [Award Number: AISG-100E-2018-005]. This research was also supported in part by the Energy Market Authority (EP award no. NRF2017EWT-EP003-061). This project was also supported by SUTD project PIE-SGP-AI-2018-01.

## Footnotes

[1]https://github.com/google/compare_gan

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
