[Supplementary Material]

In our paper, we perform an in-depth analysis to understand how SS tasks interact with the learning of the generator. We analyze the issues of SS tasks and propose to improve it with a multi-class minimax game. In this Appendix section, we provide detail information about our proofs, discussion, ablation study, network parameters and network architectures of models.

## A    Appendix: Proofs for Sections 4 and 5

**Proposition 1** (Proof.)

Let $T_k$ be the $k$-th type of transformation, and let $P_d^T$ be the distribution of the transformed real sample. This section shows the proof for optimal $C^*$. $C_k(.)$ is the $k$-th soft-max output of $C$, hence $\sum_{k=1}^{K} C_k(\mathbf{x}) = 1, \forall \mathbf{x}$. $\Psi(C)$ can be re-written as:

$$\Psi(C) = \int p_d^T(\mathbf{x}) \bigg( \sum_{k=1}^{K} p(T_k|\mathbf{x}) \log \big( C_k(\mathbf{x}) \big) \bigg) d\mathbf{x} \tag{12}$$

where $p(T_k|\mathbf{x})$ is the probability that $\mathbf{x}$ belongs to class $T_k$, which can be considered as the the $k$-th output of "ground-truth" classifier on sample $\mathbf{x}$ we expect the classifier $C$ to predict. Assume that $\Psi(C)$ has first-order derivative with respective to $C_k(\mathbf{x})$. The optimal solution of $C_k(\mathbf{x})$ can be obtained via setting this derivative equal to zero:

$$\begin{aligned}
\frac{\partial \Psi(C)}{\partial C_k(\mathbf{x})} &= \frac{\partial}{\partial C_k(\mathbf{x})} \int p_d^T(\mathbf{x}) \bigg( \sum_{k=1}^{K} p(T_k|\mathbf{x}) \log \big( C_k(\mathbf{x}) \big) \bigg) d\mathbf{x} \\
&= \frac{\partial}{\partial C_k(\mathbf{x})} \int p_d^T(\mathbf{x}) \bigg( p(T_1|\mathbf{x}) \log \big( C_1(\mathbf{x}) \big) + \sum_{k=2}^{K} p(T_k|\mathbf{x}) \log \big( C_k(\mathbf{x}) \big) \bigg) d\mathbf{x} \\
&= \frac{\partial}{\partial C_k(\mathbf{x})} \int p_d^T(\mathbf{x}) \bigg( p(T_1|\mathbf{x}) \log \big( 1 - \sum_{k=2}^{K} C_k(\mathbf{x}) \big) + \sum_{k=2}^{K} p(T_k|\mathbf{x}) \log \big( C_k(\mathbf{x}) \big) \bigg) d\mathbf{x} \\
&= p_d^T(\mathbf{x}) \bigg( \frac{p(T_k|\mathbf{x})}{C_k(\mathbf{x})} - \frac{p(T_1|\mathbf{x})}{C_1(\mathbf{x})} \bigg)
\end{aligned} \tag{13}$$

For any $k \in \{2, \ldots, K\}$, setting $\frac{\partial \Psi}{\partial C_k(\mathbf{x})} = 0$, and the value of optimal $C_k^*$ has the following form:

$$\frac{p(T_1|\mathbf{x})}{C_1^*(\mathbf{x})} = \frac{p(T_2|\mathbf{x})}{C_2^*(\mathbf{x})} = \cdots = \frac{p(T_k|\mathbf{x})}{C_K^*(\mathbf{x})} \tag{14}$$

Note that $\sum_{k=1}^{K} C_k^*(\mathbf{x}) = 1$, according to Bayes' theorem $p_d^T(\mathbf{x}) * p(T_k|\mathbf{x}) = p(T_k) * p_d^{T_k}(\mathbf{x})$, and $p(T_i) = p(T_j) = \frac{1}{K}, i, j \in [1, K]$ (the probability we apply the transformations $T_k$ for sample $\mathbf{x}$ are equal), We finally obtain the optimal $C_k^*(\mathbf{x})$ from Eq. 14: $C_k^*(\mathbf{x}) = \frac{p(T_k|\mathbf{x})}{\sum_{k=1}^{K} p(T_k|\mathbf{x})} = \frac{p_d^{T_k}(\mathbf{x})}{\sum_{k=1}^{K} p_d^{T_k}(\mathbf{x})}$.
That concludes our proof.

**Theorem 1** (Proof.) Substitute $C^*$ obtained above into $\Phi(G, C)$:

$$\Phi(G, C^*) = \int p_g^T(\mathbf{x}) \bigg[ \sum_{k=1}^{K} p(T_k|\mathbf{x}) \log \big( C_k^*(\mathbf{x}) \big) \bigg] \tag{15}$$

Substitute $C^*$ into (15) we have:

$$
\begin{aligned}
\Phi(G, C^*) &= \int p_g^T(\mathbf{x}) \sum_{k=1}^{K} \left[ p(T_k|\mathbf{x}) \log \left( C_k^*(\mathbf{x}) \right) \right] d\mathbf{x} \\
&= \int p_g^T(\mathbf{x}) \sum_{k=1}^{K} \left[ p(T_k|\mathbf{x}) \log \left( \frac{p_d^{T_k}(\mathbf{x})}{\sum_{k=1}^{K} p_d^{T_k}(\mathbf{x})} \right) d\mathbf{x} \right] \\
&= \sum_{k=1}^{K} \int \left[ p_g^T(\mathbf{x}) p(T_k|\mathbf{x}) \log \left( \frac{p_d^{T_k}(\mathbf{x})}{\sum_{k=1}^{K} p_d^{T_k}(\mathbf{x})} \right) d\mathbf{x} \right] \\
&= \sum_{k=1}^{K} \int \left[ \frac{1}{K} p_g^{T_k}(\mathbf{x}) \log \left( \frac{p_d^{T_k}(\mathbf{x})}{\sum_{k=1}^{K} p_d^{T_k}(\mathbf{x})} \right) d\mathbf{x} \right] \\
&= \frac{1}{K} \sum_{k=1}^{K} \left[ \int p_g^{T_k}(\mathbf{x}) \log \left( \frac{p_d^{T_k}(\mathbf{x})}{\sum_{k=1}^{K} p_d^{T_k}(\mathbf{x})} \right) d\mathbf{x} \right] \\
&= \frac{1}{K} \sum_{k=1}^{K} \left[ \mathbb{E}_{\mathbf{x} \sim P_g^{T_k}} \log \left( \frac{p_d^{T_k}(\mathbf{x})}{\sum_{k=1}^{K} p_d^{T_k}(\mathbf{x})} \right) \right]
\end{aligned}
\tag{16}
$$

That concludes our proof.

**Proposition 2** (Proof.) Training self-supervised task $\Psi^+(G, C)$ with minimax game is similar to previous objective, except the additional term of fake class as below:

$$
\begin{aligned}
\Psi^+(G, C) &= \mathbb{E}_{\mathbf{x} \sim P_d^T} \mathbb{E}_{T_k \sim \mathcal{T}} \log \left( C_k(\mathbf{x}) \right) + \mathbb{E}_{\mathbf{x} \sim P_g^T} \mathbb{E}_{T_k \sim \mathcal{T}} \log \left( C_{K+1}(\mathbf{x}) \right) \\
&= \int \left( p_d^T(\mathbf{x}) \sum_{i=1}^{K} p(T_k|\mathbf{x}) \log \left( C_k(\mathbf{x}) \right) + p_g^T(\mathbf{x}) \sum_{i=1}^{K} p(T_k|\mathbf{x}) \log \left( C_{K+1}(\mathbf{x}) \right) \right) d\mathbf{x}
\end{aligned}
\tag{17}
$$

Assume that $\Psi^+(G, C)$ has first-order derivative with respective to $C_k(\mathbf{x})$. The optimal $C_k^*(\mathbf{x})$ can be derived via setting derivative of $\Psi^+(G, C)$ equal to zero as follows:

$$
\begin{aligned}
&\frac{\partial \Psi^+(G, C)}{\partial C_k(\mathbf{x})} \\
&= \frac{\partial}{\partial C_k(\mathbf{x})} \int \left( p_d^T(\mathbf{x}) \sum_{i=1}^{K} p(T_k|\mathbf{x}) \log \left( C_k(\mathbf{x}) \right) + p_g^T(\mathbf{x}) \sum_{i=1}^{K} p(T_k|\mathbf{x}) \log \left( C_{K+1}(\mathbf{x}) \right) \right) d\mathbf{x} \\
&= \frac{\partial}{\partial C_k(\mathbf{x})} \int \left( p_d^T(\mathbf{x}) p(T_1|\mathbf{x}) \log \left( 1 - \sum_{k=1}^{K} C_k(\mathbf{x}) - C_{K+1}(\mathbf{x}) \right) \right. \\
&\quad \left. + p_d^T(\mathbf{x}) \sum_{k=2}^{K} p(T_k|\mathbf{x}) \log \left( C_k(\mathbf{x}) \right) + p_g^T(\mathbf{x}) \sum_{k=1}^{K} p(T_k|\mathbf{x}) \log \left( C_{K+1}(\mathbf{x}) \right) \right) d\mathbf{x}
\end{aligned}
\tag{18}
$$

Similar to above, for any $k \in \{2, \ldots, K\}$, we have the derivative $\frac{\partial \Psi^+(G,C)}{\partial C_k(\mathbf{x})}$:

$$
\frac{\partial \Psi^+(G, C)}{\partial C_k(\mathbf{x})} = p_d^T(\mathbf{x}) \left( \frac{p(T_1|\mathbf{x})}{C_1^*(\mathbf{x})} - \frac{p(T_k|\mathbf{x})}{C_k^*(\mathbf{x})} \right)
\tag{19}
$$

Setting $\frac{\partial \Psi}{\partial C_k(\mathbf{x})} = 0$, and we get optimal $C_k^*$, $k \in \{1, \ldots, K\}$:

$$\frac{p_d^T(\mathbf{x})p(T_1|\mathbf{x})}{C_1^*(\mathbf{x})} = \frac{p_d^T(\mathbf{x})p_{T_2}(\mathbf{x})}{C_2^*(\mathbf{x})} = \cdots = \frac{p_d^T(\mathbf{x})p(T_k|\mathbf{x})}{C_K^*(\mathbf{x})} = \frac{p_d^T(\mathbf{x})\sum_{k=1}^K p(T_k|\mathbf{x})}{\sum_{k=1}^K C_k^*(\mathbf{x})} \quad (20)$$

With $k = K + 1$, we obtain the derivative of $\frac{\partial \Psi^+(G,C)}{\partial C_{K+1}(\mathbf{x})}$:

$$\frac{\partial \Psi^+(G,C)}{\partial C_{K+1}(\mathbf{x})} = p_d^T(\mathbf{x})\frac{p(T_1|\mathbf{x})}{C_1^*(\mathbf{x})} - p_g^T(\mathbf{x})\frac{\sum_{k=1} Kp(T_k|\mathbf{x})}{C_{K+1}^*(\mathbf{x})} \quad (21)$$

Setting $\frac{\partial \Psi}{\partial C_{K+1}(\mathbf{x})} = 0$, and finally we get optimal $C_k^*$, $k \in \{1, \ldots, K+1\}$:

$$\frac{p_d^T(\mathbf{x})p(T_1|\mathbf{x})}{C_1^*(\mathbf{x})} = \cdots = \frac{p_d^T(\mathbf{x})p(T_k|\mathbf{x})}{C_K^*(\mathbf{x})} = \frac{p_d^T(\mathbf{x})\sum_{k=1}^K p(T_k|\mathbf{x})}{\sum_{k=1}^K C_k^*(\mathbf{x})} = \frac{p_g^T(\mathbf{x})\sum_{k=1}^K p(T_k|\mathbf{x})}{C_{K+1}^*(\mathbf{x})} \quad (22)$$

Because $\sum_{k=1}^K C_k^*(\mathbf{x}) + C_{K+1}^*(\mathbf{x}) = 1$, we finally obtain the optimal $C_k^*(x)$ from Eq. 20: $C_k^*(\mathbf{x}) = \frac{p_d^T(\mathbf{x})}{p_g^T(\mathbf{x})}\frac{p(T_k|\mathbf{x})}{\sum_{k=1}^K p(T_k|\mathbf{x})}C_{K+1}^*(\mathbf{x}) = \frac{p_d^T(\mathbf{x})}{p_g^T(\mathbf{x})}\frac{p_d^{T_k}(\mathbf{x})}{\sum_{k=1}^K p_d^{T_k}(\mathbf{x})}C_{K+1}^*(\mathbf{x})$. That concludes the proof.

**Theorem 2** (Proof.) Substitute optimal $C^*$ obtained above into $\Phi^+(G,C)$:

$$\Phi^+(G,C^*) = \left( \mathbb{E}_{\mathbf{x} \sim P_g^T} \sum_{k=1}^K p(T_k|\mathbf{x}) \log \left( C_k^*(\mathbf{x}) \right) - \mathbb{E}_{\mathbf{x} \sim P_g^T} \sum_{k=1}^K p(T_k|\mathbf{x}) \log \left( C_{K+1}^*(\mathbf{x}) \right) \right) \quad (23)$$

The first term can be written as:

$$\mathbb{E}_{\mathbf{x} \sim P_g^T} \sum_{k=1}^K p(T_k|\mathbf{x}) \log(C_k^*(\mathbf{x}))$$

$$= \mathbb{E}_{\mathbf{x} \sim P_g^T}\left[ \sum_{k=1}^K p(T_k|\mathbf{x}) \log \left( \frac{p_d^T(\mathbf{x})}{p_g^T(\mathbf{x})}\frac{p(T_k|\mathbf{x})}{\sum_{k=1}^K p(T_k|\mathbf{x})}C_{K+1}^*(\mathbf{x}) \right) \right]$$

$$= \mathbb{E}_{\mathbf{x} \sim P_g^T}\left[ \sum_{k=1}^K p(T_k|\mathbf{x}) \log \left( C_{K+1}^*(\mathbf{x}) \right) + \log \left( \frac{p_d^T(\mathbf{x})}{p_g^T(\mathbf{x})} \right) + \log \left( \frac{p(T_k|\mathbf{x})}{\sum_{k=1}^K p(T_k|\mathbf{x})} \right) \right]$$

$$= \mathbb{E}_{\mathbf{x} \sim P_g^T}\left[ \sum_{k=1}^K p(T_k|\mathbf{x}) \log \left( C_{K+1}^*(\mathbf{x}) \right) \right] + \mathbb{E}_{\mathbf{x} \sim P_g^T}\left[ \sum_{k=1}^K p(T_k|\mathbf{x}) \log \left( \frac{p_d^T(\mathbf{x})}{p_g^T(\mathbf{x})} \right) \right]$$

$$+ \mathbb{E}_{\mathbf{x} \sim P_g^T}\left[ \sum_{k=1}^K p(T_k|\mathbf{x}) \log \left( \frac{p(T_k|\mathbf{x})}{\sum_{k=1}^K p(T_k|\mathbf{x})} \right) \right] \quad (24)$$

$$= \mathbb{E}_{\mathbf{x} \sim P_g^T}\left[ \sum_{k=1}^K p(T_k|\mathbf{x}) \log \left( C_{K+1}^*(\mathbf{x}) \right) \right] + \frac{1}{K}\left[ \sum_{k=1}^K \int p_g^{T_k}(\mathbf{x}) \log \left( \frac{p_d^{T_k}(\mathbf{x})}{p_g^{T_k}(\mathbf{x})} \right) d\mathbf{x} \right]$$

$$+ \mathbb{E}_{\mathbf{x} \sim P_g^T}\left[ \sum_{k=1}^K p(T_k|\mathbf{x}) \log \left( \frac{p(T_k|\mathbf{x}) * p_d^T(\mathbf{x})}{\sum_{k=1}^K p(T_k|\mathbf{x}) * p_d^T(\mathbf{x})} \right) \right]$$

$$= \mathbb{E}_{\mathbf{x} \sim P_g^T}\left[ \sum_{k=1}^K p(T_k|\mathbf{x}) \log \left( C_{K+1}^*(\mathbf{x}) \right) \right] - \frac{1}{K}\left[ \sum_{k=1}^K \mathrm{KL}(P_g^{T_k} || P_d^{T_k}) \right]$$

$$+ \frac{1}{K}\sum_{k=1}^K \left[ \mathbb{E}_{\mathbf{x} \sim P_g^{T_k}} \log \left( \frac{p_d^{T_k}(\mathbf{x})}{\sum_{k=1}^K p_d^{T_k}(\mathbf{x})} \right) \right]$$

With the note that $p_d^T(\mathbf{x}) * p(T_k|\mathbf{x}) = p(T_k) * p_d^{T_k}(\mathbf{x}) = \frac{1}{K}p_d^{T_k}(\mathbf{x})$ and $p_g^T(\mathbf{x}) * p(T_k|\mathbf{x}) = p(T_k) * p_g^{T_k}(\mathbf{x}) = \frac{1}{K}p_g^{T_k}(\mathbf{x})$. Moving the first term of Eq. 24 from the right side to left side, it concludes the proof.

**Theorem 3** *KL divergence is invariant to affine transform.*

*Proofs.* Let $\mathbf{x} \in \mathbb{R}^{n \times 1}$ be a random variable. $p_x(\mathbf{x})$ is a distribution defined on $\mathbf{x}$. Let $T$ be an affine transform, i.e., $T(\mathbf{x}) = \boldsymbol{A}\mathbf{x} + \boldsymbol{b}$, where $\boldsymbol{A} \in \mathbb{R}^{n \times n}$ is a full rank matrix and $\boldsymbol{b} \in \mathbb{R}^{n \times 1}$. Then for a random variable $\mathbf{y} = T(\mathbf{x}) = \boldsymbol{A}\mathbf{x} + \boldsymbol{b}$, $p_y(\mathbf{y}) = |\boldsymbol{J}|p_x\left(T^{-1}(\mathbf{y})\right)$, where $\boldsymbol{J}$ is the Jacobian matrix, with its $(i,j)$-th entry defined as:

$$\boldsymbol{J}_{i,j} = \frac{\partial \mathbf{x}_i}{\partial \mathbf{y}_j} \tag{25}$$

Obviously, $\boldsymbol{J} = \boldsymbol{A}^{-1}$. Then we have $p_y(\mathbf{y}) = |\boldsymbol{A}^{-1}|p_x\left(T^{-1}(\mathbf{y})\right)$.

Let $p_{x_1}(\mathbf{x})$ and $p_{x_2}(\mathbf{x})$ are two distributions defined on $\mathbf{x}$. Then let $p_{y_1}(\mathbf{y})$ and $p_{y_2}(\mathbf{y})$ be the corresponding distributions defined on $\mathbf{y}$. Then we have $p_{y_1}(\mathbf{y}) = |\boldsymbol{A}^{-1}|p_{x_1}\left(T^{-1}(\mathbf{y})\right)$ and $p_{y_2}(\mathbf{y}) = |\boldsymbol{A}^{-1}|p_{x_2}\left(T^{-1}(\mathbf{y})\right)$.

Using the definition of the KL divergence between $p_{y_1}$ and $p_{y_2}$, we have:

$$\mathrm{KL}(p_{y_1}||p_{y_2}) = \int p_{y_1}(\mathbf{y}) \log \frac{p_{y_1}(\mathbf{y})}{p_{y_2}(\mathbf{y})} d\mathbf{y} \tag{26}$$

$$= \int |\boldsymbol{A}^{-1}|p_{x_1}\left(T^{-1}(\mathbf{y})\right) \log \frac{|\boldsymbol{A}^{-1}|p_{x_1}(T^{-1}(\mathbf{y}))}{|\boldsymbol{A}^{-1}|p_{x_2}\left(T^{-1}(\mathbf{y})\right)} d\mathbf{y} \tag{27}$$

$$= \int |\boldsymbol{A}^{-1}|p_{x_1}\left(T^{-1}(\mathbf{y})\right) \log \frac{p_{x_1}\left(T^{-1}(\mathbf{y})\right)}{p_{x_2}\left(T^{-1}(\mathbf{y})\right)} d\mathbf{y} \tag{28}$$

As $\mathbf{x} = T^{-1}(\mathbf{y})$, then we have:

$$\mathrm{KL}(p_{y_1}||p_{y_2}) = \int |\boldsymbol{A}^{-1}|p_{x_1}\left(T^{-1}(\mathbf{y})\right) \log \frac{p_{x_1}\left(T^{-1}(\mathbf{y})\right)}{p_{x_2}\left(T^{-1}(\mathbf{y})\right)} d\mathbf{y} \tag{29}$$

$$= \int |\boldsymbol{A}^{-1}|p_{x_1}(\mathbf{x}) \log \frac{p_{x_1}(\mathbf{x})}{p_{x_2}(\mathbf{x})} d\mathbf{y} \tag{30}$$

According to the property of multiple integral, we have:

$$\mathrm{KL}(p_{y_1}||p_{y_2}) = \int |\boldsymbol{A}^{-1}|p_{x_1}(\mathbf{x}) \log \frac{p_{x_1}(\mathbf{x})}{p_{x_2}(\mathbf{x})} |\boldsymbol{A}|d\mathbf{x} \tag{31}$$

$$= \int p_{x_1}(\mathbf{x}) \log \frac{p_{x_1}(\mathbf{x})}{p_{x_2}(\mathbf{x})} d\mathbf{x} \tag{32}$$

$$= \mathrm{KL}(p_{x_1}||p_{x_2}) \tag{33}$$

It concludes our proof.

**Corollary 1** *KL divergence between real and fake distributions is equal to that of rotated real and rotated fake distributions by $T_k$: $\mathrm{KL}(P_g^{T_k}||P_d^{T_k}) = \mathrm{KL}(P_g||P_d), k \in [1:K]$*

Note that we apply the above theorem of invariance of KL, with $p_{x_1}, p_{x_2}$ being $P_g, P_d$ respectively, and image rotation $T_k$ as the transform.

## A.1 Implementation

Here, we discuss details of our implementation. For the SS tasks, we follow the geometric transformation of [10] to argument images and compute pseudo labels. It is simple yet effective and currently the state-of-the-art in self-supervised tasks. In particular, we train discriminator to recognize the 2D rotations which were applied to the input image. We rotate the input image with $K = 4$ rotations $(0°, 90°, 180°, 270°)$ and assign them the pseudo-labels from 1 to $K$.

To implement our model, the GAN objectives for discriminator and generator can be the ones in original GAN by Goodfellow et al. [12], or other variants. In our work, we conduct experiments to show improvements with two baseline models: original SSGAN [4] and DistGAN [42].

We integrate SS tasks into Dist-GAN [42] and conduct study with this baseline. In our experiments, we observe that Dist-GAN has good convergence property and this is important for our ablation study.

$$
\min_G \mathcal{V}(D, C, G) = \mathcal{V}(D, G)
$$
$$
+ \lambda_g \left\| \left( \mathbb{E}_{\mathbf{x} \sim P_g^T} \mathbb{E}_{T_k \sim \mathcal{T}} \log \left( C_k(\mathbf{x}) \right) - \mathbb{E}_{\mathbf{x} \sim P_g^T} \mathbb{E}_{T_k \sim \mathcal{T}} \log \left( C_{K+1}(\mathbf{x}) \right) \right) \right.
$$
$$
\left. - \left( \mathbb{E}_{\mathbf{x} \sim P_d^T} \mathbb{E}_{T_k \sim \mathcal{T}} \log \left( C_k(\mathbf{x}) \right) - \mathbb{E}_{\mathbf{x} \sim P_d^T} \mathbb{E}_{T_k \sim \mathcal{T}} \log \left( C_{K+1}(\mathbf{x}) \right) \right) \right\|
\tag{34}
$$

Second, in practice, achieving equilibrium point for optimal D, G, C is difficult. Therefore, inspired by [42], we propose the new generator objective to improve Eq. 9 as written in Eq. 34. It couples the convergence of $\Phi^+(G, C)$ and $\Psi^+(G, C)$ that allows the learning is more stable. Our intuition is that if generator distribution is similar to the real distribution, the classification performance on its transformed fake samples should be similar to that of those from real samples. Therefore, we propose to match the self-supervised tasks of real and fake samples to train the generator. In other words, if real and fake samples are from similar distributions, the same tasks applied for real and fake samples should have resulted in similar behaviors. In particular, given the cross-entropy loss computed on real samples, we train the generator to create samples that are able to match this loss. Here, we use $\ell_1$-norm for the $\Phi^+(G, C)$ and $\mathcal{V}(D, G)$ is the objective of GAN task [42]. In our implementation, we randomly select a geometric transformation $T_k$ for each data sample when training the discriminator. And the same $T_k$ are applied for generated samples when matching the self-supervised tasks to train the generator.

For this objective of generator, similar to Eq. 24, we have:

$$
\mathbb{E}_{\mathbf{x} \sim P_d^T} \left[ \sum_{k=1}^K p(T_k | \mathbf{x}) \log \left( C_k^*(\mathbf{x}) \right) \right]
$$
$$
= \mathbb{E}_{\mathbf{x} \sim P_d^T} \left[ \sum_{k=1}^K p(T_k | \mathbf{x}) \log \left( C_{K+1}^*(\mathbf{x}) \right) \right] + \left[ \sum_{k=1}^K \mathrm{KL}(P_d^{T_k} || P_g^{T_k}) \right]
\tag{35}
$$
$$
+ \sum_{k=1}^K \left[ \mathbb{E}_{\mathbf{x} \sim P_d^{T_k}} \log \left( \frac{p_d^{T_k}(\mathbf{x})}{\sum_{k=1}^K p_d^{T_k}(\mathbf{x})} \right) \right]
$$

The objective of Eq. 34 can be re-written as:

$$
* = \left\| \sum_{k=1}^K \left( \mathrm{KL}(P_d^{T_k} || P_g^{T_k}) + \mathrm{KL}(P_g^{T_k} || P_d^{T_k}) \right) \right.
$$
$$
\left. + \sum_{k=1}^K \mathbb{E}_{\mathbf{x} \sim P_g^{T_k}} \log \left( \frac{p_d^{T_k}(\mathbf{x})}{\sum_{k=1}^K p_d^{T_k}(\mathbf{x})} \right) - \sum_{k=1}^K \mathbb{E}_{\mathbf{x} \sim P_d^{T_k}} \log \left( \frac{p_d^{T_k}(\mathbf{x})}{\sum_{k=1}^K p_d^{T_k}(\mathbf{x})} \right) \right\| \geq 0
\tag{36}
$$

$P_g = P_d$ is the solution that minimizes Eq. 36. In practice, we found that this is stable. It is due to the stability of symmetric KL divergence (forward KL and inverse KL).

Figure 5: From left to right: Real samples, argument real samples by rotation, mixed argument real and fake samples, and generated images of CIFAR-10.

# B    Appendix: Experiments

## B.1    Details of experiment setup

In our experiments, FID is computed every 10K iterations in training and visualized with the smoothening windows of 5. The latent dimension is $d_z = 128$ and mini-batch size is 64 for our all experiments. We visualize losses and FID scores in several figures. In these figures, the horizontal axis is the number of training iterations, and the vertical axis is either the loss and FID score. We compute the negative discriminator/classifier value function for the visualization. We investigate the improvements of our proposed techniques on two baseline models:

Dist-GAN [42]: We use Dist-GAN implemented with three network architectures: DCGAN, CNN in SN-GAN and ResNet. We use standard "log" loss for DCGAN architecture, and with "hinge" loss SN-GAN (the CNN network as in SN-GAN [31]) and ResNet architectures. We use "hinge" loss for SN-GAN and ResNet because it attains better performance than standard "log" loss as shown in [31]. We train models using Adam optimizer with learning rate lr = 0.0002, $\beta_1 = 0.5$, $\beta_2 = 0.9$ for DCGAN and SN-GAN architectures and $\beta_1 = 0.0$, $\beta_2 = 0.9$ for ResNet architecture [13]. **If not precisely mentioned, it means Dist-GAN is used for the experiments.**

Figure 6: From left to right: Real samples, argument real samples by rotation, mixed argument real and fake samples, and generated images of STL-10.

SSGAN: We were unable to reproduce results as reported in the original paper with this code[2], although we have followed the best parameter settings of the paper and communicated with authors of SSGAN regarding the issues. We achieve best results with another setting (spectral norm, $\lambda = 0, \mathrm{d}_{\mathrm{iter}} = 10, \beta_1 = 0.5, \beta_2 = 0.999$). We use this setting as the baseline and compare to the one using our proposed SS tasks instead of the original SS tasks.

### B.2 Ablation study SS in Discriminator and Generator Learning for the original SS proposed in [4]

In this experiment, we analyze original SS tasks proposed in [4] to understand the effect of self-supervised tasks. We aim to provide empirical observation of how the $\Psi(C)$ contributes to the discriminator via changing $\lambda_d$ with fixed $\lambda_g = 0$. Experiments are on CIFAR-10 dataset using small DCGAN architecture. For implementation, they are integrated into the discriminator of the baseline model, Dist-GAN [42] as mentioned above. Through the experiment, we confirm that the contribution of $\Psi(C)$ is important in Dist-GAN model. We should set the $\lambda_d$ attain the good trade-off between GAN task and SS task because increasing $\lambda_d$ is not helpful. The SS task with $\lambda_d = 1.0$ is good for Dist-GAN model, which is also discussed in [4] with SN-GAN model [31].

Figure 7: The ablation study of SS task $\Psi(C)$ as proposed in [4]. We analyze its effect via $\lambda_d$ fine-tuning, $\lambda_g = 0.0$. (a) The discriminator losses of SS task, (b) The discriminator losses of GAN task, (c) the feature representation quality and (d) FID scores. With $\lambda_d = 7.0$ for SS task, the model becomes seriously collapsed with FID > 100. Experiments are conducted with the baseline model, Dist-GAN. (Best view in color).

The results in Fig. 7 illustrate the effects of $\Psi(C)$ to GAN with different values of $\lambda_d$. Fig. 7a represents the losses of the SS task of the discriminator. It shows that in most cases, the larger $\lambda_d$ lead to faster $\Psi(C)$ loss converges. However, when $\lambda_d > 1.0$, the FID is not improved. We observe that once $\lambda_d$ is higher, the loss of GAN task is dominated by the SS tasks. When $\lambda_d$ is too high, (e.g., $\lambda_d = 7.0$), GAN loss is almost unchanged about first 10K iterations (Fig. 7b) in early iterations and the model gets collapsed. This can be explained as follows. When the discriminator improvement is slow due to the strong dominance of $\Psi(C)$, the learning of the generator faster. This serious unbalance easily leads to the collapsed generator and the learning of generator gets stuck thereafter. When the GAN loss is strongly dominated by SS loss, the loss of GAN is saturated.

To understand deeper, we evaluate the representation qualities of the intermediate layers of the discriminator as in [4] in this experiment. Given the above pre-trained discriminators, we compute features of train and test sets of CIFAR-10 via its last convolution layer. We evaluate the classification performance as training logistic regression on these features and measure with top-1 accuracy. We follow the experimental setup of parameters as in [4]. The result (Fig. 7c) that as $\lambda_d \geq 1.0$, the accuracy is also similar, except for the case $\lambda = 7.0$, the quality of feature is slightly worse but not too significant (although the GAN model is collapsed). It means increasing $\lambda_d$ does not necessarily improve the feature representation quality of the discriminator.

Overall, with Dist-GAN as baseline, we observe that using the original SS tasks with $\lambda_d = 1.0$ provide considerable improvement, and the results suggest that $\lambda_d$ should not be too high, but instead the one that provides a good trade-off between GAN and SS tasks.

Figure 8: The ablation studies with (a) SSGAN$^+$ using $\lambda_g$ and ours (SSGAN$^+$ + MS) (b) Dist-GAN with $\Phi(C)$ and $\Psi(G,C)$ as fine-tuning $\lambda_g$, fixed $\lambda_d = 1.0$. (c) Our model (Dist-GAN + MS) with $\Psi^+(G,C)$ and $\Phi^+(G,C)$ with $\lambda_g = 0.1$ and $\lambda_d = 1.0$. Experiments are with CIFAR-10 dataset. (Best view in color)

Figure 9: Our model (Dist-GAN + MS) with (a) with fine-tuning $\lambda_g$, fixed $\lambda_d = 1.0$. (b) fine-tuning $\lambda_d$, fixed $\lambda_g = 0.1$. The baseline is Dist-GAN model, and we use DCGAN architecture. (Best view in color)

### B.2.1 SS task in Generator Learning

We continue to investigate the effects of $\lambda_g$ with fixed $\lambda_d = 1.0$ for the SS tasks proposed in [4]. The experimental setup is similar to the previous one. The result represented in Fig. 8a show that $\lambda_g > 0$ still improves the baseline model, but higher than the case of $\lambda_g = 0$. Note that [4] does not report result with $\lambda_g = 0.0$.

Following our discussion on Theorem 1, applying $\Phi(G,C)$ as proposed in [4] does not support the matching between the generator and data distributions. From these experiments, we observe that the generator and discriminator are unable to reach optimal points, and using large $\lambda_g$ degrades the quality of GAN task, and even leads to mode collapse. For example, as $\lambda_g$ increases (eg. $\lambda_g = 0.3$), it seriously hurts the quality of GAN task of the generator.

In addition, we verify with original code of SSGAN [4] on CIFAR-10 using our best setting mentioned above. Fig. 8a confirms that with our best setting $\lambda_g = 0.01$ and $\lambda_g = 0.2$ achieve similar FID and increasing $\lambda_g = 0.5$ degrades its performance, which is consistent to our analysis. In the same figure, when we use our proposed MS, FID is improved.

### B.3 Ablation study $(\lambda_d, \lambda_g)$ with DCGAN for our proposed method

We first change the $\lambda_g$ according $\lambda_d = 1.0$ (Fig. 9a). With minimax game, the result suggests that $\lambda_g = 0.1$ is the best for DCGAN architecture. Then, we seek $\lambda_d$ with this $\lambda_g = 0.1$ as shown in Fig. 9b. Interestingly, now the best $\lambda_d = 4.0$, which is higher than $\lambda_d = 1.0$ of the original SS (the best with the original SS; Fig. 7d). This suggests that using our proposed mini-max game based SS enable larger range of $\lambda_d$ with stable performance.

Figure 10: Understanding the effects of MS tasks (our proposed self-supervised tasks), by fine-tuning $\lambda_g$ (first row) $\lambda_d$ (second row) for CIFAR-10 and STL-10 with other architectures. From left to right: SN-GAN for CIFAR-10, ResNet for CIFAR-10, SN-GAN for STL-10 and ResNet for STL-10. SN-GAN architecture referred as CNN architectures used in SN-GAN [31].

## B.4 Ablation study of our proposed method with SN-GAN and ResNet architectures

The detail of the ablation study of $\lambda_d$ and $\lambda_g$ for our proposed SS tasks using SN-GAN and ResNet architectures are shown in Fig. 10.

# C Appendix: Network architectures

## C.1 DCGAN architecture

Our DCGAN architecture, which is used for ablation studies on CIFAR-10, are presented in Table. 4.

Table 4: Our DCGAN architecture is similar to [37] but the smaller number of feature maps (D = 64) to be more efficient for our ablation study on CIFAR-10. The Encoder is the mirror of the Generator. Slopes of lReLU functions are set to 0.2. $\mathcal{U}(0,1)$ is the uniform distribution.

| RGB image $x \in \mathbb{R}^{M \times M \times 3}$ |
| --- |
| 5×5, stride=2 conv. 1 × D lReLU |
| 5×5, stride=2 conv. BN 2 × D lReLU |
| 5×5, stride=2 conv. BN 4 × D lReLU |
| 5×5, stride=2 conv. BN 8 × D lReLU |
| dense → 1, dense → 5 (two heads) |

| RGB image $x \in \mathbb{R}^{M \times M \times 3}$ |
| --- |
| 5×5, stride=2 conv. 1 × D ReLU |
| 5×5, stride=2 conv. BN 2 × D ReLU |
| 5×5, stride=2 conv. BN 4 × D ReLU |
| 5×5, stride=2 conv. BN 8 × D ReLU |
| dense → 128 |

(a) Encoder, $M = 32$ for CIFAR-10

| $z \in \mathbb{R}^{128} \sim \mathcal{U}(0,1)$ |
| --- |
| dense → 2 × 2 × 8 × D |
| 5×5, stride=2 deconv. BN 4 × D ReLU |
| 5×5, stride=2 deconv. BN 2 × D ReLU |
| 5×5, stride=2 deconv. BN 1 × D ReLU |
| 5×5, stride=2 deconv. 3 Sigmoid |

(b) Generator for CIFAR-10

(c) Discriminator, $M = 32$ for CIFAR-10. Two heads for the real/fake discriminator and multi-class classifier.

## C.2 SNGAN architecture

Our SN-GAN architecture referred as CNN architectures of [31] for CIFAR-10 and STL-10 datasets are presented in Table. 5.

## C.3 ResNet architecture

Our ResNet architectures for CIFAR-10 and STL-10 are presented in Table. 6 and Table. 7.

Table 5: Encoder, generator, and discriminator of standard CNN architectures for CIFAR-10 and STL-10 used in our experiments. We use similar architectures as ones in [31]. The Encoder is the mirror of the Generator. Slopes of lReLU functions are set to 0.1. $\mathcal{U}(0,1)$ is the uniform distribution.

| RGB image $x \in \mathbb{R}^{M \times M \times 3}$ |
| --- |
| 3×3, stride=1 conv 64 lReLU |
| 4×4, stride=2 conv 64 lReLU |
| 3×3, stride=1 conv 128 lReLU |
| 4×4, stride=2 conv 128 lReLU |
| 3×3, stride=1 conv 256 lReLU |
| 4×4, stride=2 conv 256 lReLU |
| 3×3, stride=1 conv. 512 lReLU |
| dense → 1, dense → 5 (two heads) |

| RGB image $x \in \mathbb{R}^{M \times M \times 3}$ |
| --- |
| 3×3, stride=1 conv. 64 |
| 4×4, stride=2 conv. BN 128 ReLU |
| 4×4, stride=2 conv. BN 256 ReLU |
| 4×4, stride=2 conv. BN 512 ReLU |
| dense → 128 |

| $z \in \mathbb{R}^{128} \sim \mathcal{U}(0,1)$ |
| --- |
| dense → $M_g \times M_g \times 512$ |
| 4×4, stride=2 deconv. BN 256 ReLU |
| 4×4, stride=2 deconv. BN 128 ReLU |
| 4×4, stride=2 deconv. BN 64 ReLU |
| 3×3, stride=1 conv. 3 Sigmoid |

(a) Encoder, $M = 32$ for CIFAR-10, and $M = 48$ for STL-10

(b) Generator, $M_g = 4$ for CIFAR-10, and $M_g = 6$ for STL-10

(c) Discriminator, $M = 32$ for CIFAR-10, and $M = 48$ for STL-10. Two heads for the real/fake discriminator and multi-class classifier.

Table 6: ResNet architecture for CIFAR10 dataset. The Encoder is the mirror of the Generator. We use similar architectures and ResBlock to the ones used in [31]. $\mathcal{U}(0,1)$ is the uniform distribution.

| RGB image $x \in \mathbb{R}^{32 \times 32 \times 3}$ |
| --- |
| 3×3 stride=1, conv. 256 |
| ResBlock down 256 |
| ResBlock down 256 |
| ResBlock down 256 |
| dense → 128 |

(a) Encoder

| $z \in \mathbb{R}^{128} \sim \mathcal{U}(0,1)$ |
| --- |
| dense, $4 \times 4 \times 256$ |
| ResBlock up 256 |
| ResBlock up 256 |
| ResBlock up 256 |
| BN, ReLU, 3×3 conv, 3 Sigmoid |

(b) Generator

| RGB image $x \in \mathbb{R}^{32 \times 32 \times 3}$ |
| --- |
| ResBlock down 128 |
| ResBlock down 128 |
| ResBlock 128 |
| ResBlock 128 |
| ReLU |
| Global sum pooling |
| dense → 1, dense → 5 (two heads) |

(c) Discriminator. Two heads for the real/fake discriminator and multi-class classifier.

Table 7: ResNet architecture for STL-10 dataset. The Encoder is the mirror of the Generator. We use similar architectures and ResBlock to the ones used in [31]. $\mathcal{U}(0,1)$ is the uniform distribution.

| RGB image $x \in \mathbb{R}^{48 \times 48 \times 3}$ |
| --- |
| 3×3 stride=1, conv. 64 |
| ResBlock down 128 |
| ResBlock down 256 |
| ResBlock down 512 |
| dense → 128 |

(a) Encoder

| $z \in \mathbb{R}^{128} \sim \mathcal{U}(0,1)$ |
| --- |
| dense, $6 \times 6 \times 512$ |
| ResBlock up 256 |
| ResBlock up 128 |
| ResBlock up 64 |
| BN, ReLU, 3×3 conv, 3 Sigmoid |

(b) Generator

| RGB image $x \in \mathbb{R}^{48 \times 48 \times 3}$ |
| --- |
| ResBlock down 64 |
| ResBlock down 128 |
| ResBlock down 256 |
| ResBlock down 512 |
| ResBlock 1024 |
| ReLU |
| Global sum pooling |
| dense → 1, dense → 5 (two heads) |

(c) Discriminator. Two heads for the real/fake discriminator and multi-class classifier.

## Footnotes

[2]https://github.com/google/compare_gan