[Reviews · NeurIPS 2019]

Reviewer 1



Originality: The method is relatively new although it is similar to some conditional GAN works in the literature. The main idea is the analysis showing the limitations of prior GAN+SSL work and in proposing a scheme with better chances of succeeding (at least theoretically). Then experiments show that there is an improvement. It would be good to show more the analogies to prior conditional GAN work, and this would not hurt the contribution, rather it would better clarify its context and provide more links to practitioners (who could better understand it). ++++++++++++++++++++++ Quality: There are some inconsistencies that I would like the authors to clarify. Basically, the minimax game should use the same cost function for the optimization of the discriminator, the generator and the classifier. However, as it is evident from the general formulation in eq 1 and 2, this is not the case. It would be good to clarify that this is not a standard minimax optimization. If we ignore this inconsistency in the optimization objective function, then I still have some considerations that I would like the authors to respond to. To my understanding the proposed method (eq 11) is also probably not the ultimate solution, as the optimal solution of eq 11 is a tradeoff between the ideal loss (the KL term) and the additional SSL term. Indeed, after eq 11 one can continue the calculations and get to the final optimization function for the generator G. As a first step, the optimal D in eq 8 is the usual one as the second term does not depend on D. Thus, we can plug in the expression for D(x) = pd(x)/(pd(x)+pg(x)) and the expression in eq 11 to eq 9 to calculate the cost function in G. If lambda_g = 1 (and my calculations are not wrong), this expression simplifies in a KL divergence between (pd+pg)/2 and pd summed to a constant and the second term in eq 11. This means 2 things: 1) the new objective function in G is not symmetric in the distributions pd and pg (as in the original formulation); 2) because of the second term, the solution in G will be a tradeoff between the GAN objective and the classifier objective. A tradeoff could hurt both objectives and in particular be worse for the GAN objective, which is the one of interest. Could the authors clarify if my derivation is wrong or if my conclusions are wrong with some clear mathematical/logical explanation? In general, I would make theorems out of major results and some of the theorems in the paper are basic intermediate calculations. For example, Theorem 1 is quite trivial and I would not make a theorem out of a simple substitution. ++++++++++++++++++++++ Clarity: The paper is clearly written for most part. It clearly has a space problem as it needs space to write down equations and theorem proofs. This is helped by the supplementary material, but I have the impression that more could be written in the theoretical analysis (eg see my comments above on continuing the derivation until one gets to the final optimization in G). The work is probably non trivial to reproduce so sharing the code would be quite important. There are some typos and grammar errors, but they are not too serious. It would certainly help the reader if the authors revised the writing or had a native English speaker revise the text. ++++++++++++++++++++++ Significance: The results show that the proposed method is better than [4]. It is always difficult to judge whether this is due to the proposed change or instead to a better tuning and engineering. The authors mention that they also searched parameters for the competing method but they could not replicate the published performance. This makes these experimental evaluations difficult to assess. +++++++++++++++++++++ I read all the reviews and the rebuttal. We discussed briefly the rebuttal and I decided to keep my score.

Reviewer 2



Overall the paper is well-written and easy to understand. I actually like the paper a lot. The logics are also clear: Discovering the limitation of the previous methods using both theoretical analysis and toy example. Then, proposed the solution (in a simple way) that works better. The only concern is about the empirical experiments which are done only in the simple dataset. If the authors provide the results in more complex datasets (such as Imagenet and CIFAR-100), it would improve the quality of this paper 1. Self-supervised signal - The proposed model and the previous models usually use the rotation as the self-supervised signal. - Are there any other signals that can be used for the self-supervised signal? - Because the rotation can be only used for image dataset and cannot be extended to the other domains such as time-series or tabular data. 2. Figure 3 - Line 167 - 180: The author claims that the previous method has the possibility to mode collapse for generating the samples whose rotations are easily captured by the classifier. - It would be good if the author also shows the results of the proposed model and claims that the proposed method has a small mode collapse problem. - The point is that the author can use the toy example not only showing the limitation of the previous method, but also show the superiority of the proposed method. Currently, I can understand that there is a problem in the previous method; however, it is not clear whether the proposed method can solve this problem in the same setting. 3. Section 5 - It seems like the additional losses can be interpreted as another discriminator that distinguish the rotated fake image and the rotated real image. - In that case, why the authors put this in the classifier? Can we make another discriminator to distinguish the real rotated image and fake rotated image separately? In this case, the separate discriminator has higher flexibility. - In the theoretical analysis, the author also mentioned that this minimizes the KL between rotated real and fake images. - It would be good to show the results of this as well in order to support the proposed method in comparison to other possible models. 4. Datasets - The author provided the results on fairly small and less complex datasets (CIFAR-10 and STL-10 datasets). - If the authors provide the results on more complex datasets (with more labels (means diverse)) such as CIFAR-100 and Imagenet, it will improve the quality of the paper.

Reviewer 3



This paper provides an in-depth discussion on the Self-supervised (SS) learning mechanism in GAN training. The theoretical analysis reveals that the existing SS design indulges mode-collapsed generators. This work presents an improved SS task for GAN, which is shown to achieve better convergence property in both theoretical and empirical analysis. The paper is well-written. It is easy to follow as the logic presents both the motivation and the solution in a clear way. The analysis regarding the interaction between the SS task and the learning of generator provides insights for other design of GAN training methods. Empirical results are shown both in the main content and the appendix. Below are some confusions that the author is better to clarify in the rebuttal. 1. Figure.3 is hard to understand based on the content in Page 5, Line 151-157. The author may better explain what Figure.3 (a) and (b) shows here, clearly. 2. The paper directly adopts many reported results of the baselines in other papers, e.g. Table 1. Are these methods compared under the same setting? Moreover, many results are missing here without explaining why. I encourage the author to clarify this. 3. As the improved version focuses on tackling the mode-collapse problem here. It is better to presents comparison on some criteria, e.g. the number of modes covered, in the main body of the paper.

[Author Response · NeurIPS 2019]

1 We sincerely thank Reviewers and Area Chairs for valuable feedback.

2 **Reviewer 1** • **1. The optimization function for generator:** We really appreciate Reviewer's suggestion. We have
3 checked Reviewer's derivation carefully. We humbly point out that there is a small sign mistake in the derivation:
4 $\lambda_g \Phi^+(G,C^*)$ was used in Eq. 9, but the correct one is $-\lambda_g \Phi^+(G,C^*)$. We have derived generator optimization. Using
5 $\sum_{k=1}^{K} \text{KL}(P_g^{T_k}||P_d^{T_k}) = K \cdot \text{KL}(P_g||P_d)$, and let $\mathcal{V}_\Phi(\mathbf{x}) = \sum_{k=1}^{K} p^{T_k}(\mathbf{x}) \log(\frac{p^{T_k}(\mathbf{x})}{\sum_{k=1}^{K} p^{T_k}(\mathbf{x})})$, we substitute Eq. 11 into Eq. 9:

$$\min_G \mathcal{V}(D^*, C^*, G) = \mathcal{V}(D^*, G) - \lambda_g \left( - K \cdot \text{KL}(P_g||P_d) + \mathbb{E}_{\mathbf{x} \sim P_g^T} \mathcal{V}_\Phi(\mathbf{x}) \right)$$

$$= \int_x p_d(\mathbf{x}) \log \frac{p_d(\mathbf{x})}{p_d(\mathbf{x}) + p_q(\mathbf{x})} dx + \int_x p_g(\mathbf{x}) \log \frac{p_g(\mathbf{x})}{p_d(\mathbf{x}) + p_g(\mathbf{x})} dx + K\lambda_g \int_x p_g(\mathbf{x}) \log \frac{p_g(\mathbf{x})}{p_d(\mathbf{x})} dx - \lambda_g \mathbb{E}_{\mathbf{x} \sim P_g^T} \mathcal{V}_\Phi(\mathbf{x})$$

6 Even we assume $K\lambda_g=1$, it is not easy to simplify $\mathcal{V}(D^*,C^*,G)$. We will further investigate.

7 • 2. Reviewer's comment on analogies to conditional GAN is thought-provoking. Previous work [4] and ours have
8 regarded self-supervised (SS) GAN as unconditional GAN. But Reviewer is correct that SS GAN can also be regarded
9 as conditional GAN with pseudo-labels. Following Reviewer's comment, we will discuss more analogies in the paper.
10 In particular, AC-GAN [30] with pseudo-labels is quite similar to SS-GAN proposed in [4], but deeper analysis is
11 needed to understand the impact of replacing class labels in the original AC-GAN with pseudo-labels.

12 • 3. We will clarify minimax optimization and improve presentation (theorems, typos). We will definitely share code.

13 **Reviewer 2** • **1. Self-supervised (SS) signals:** We thank Reviewer for feedback. Potentially, Contrastive Predictive
14 Coding [*Aaron van den Oord et al. 2018*] can be SS signals for time-series.

19 Figure A: Our proposed SS loss $-\Phi^+(G,C)$. Red:
20 Good generator; Green: Mode-collapsed generators
21 that correspond to different classes of CIFAR-10, as
22 explained in Sec.4.

• 2. We follow Reviewer's suggestion to use the same toy example to show improvement of our proposed method. The setup is the same as in our paper Sec.4 L167-180, except that now we replace models/cost functions of [4] with our proposed ones, i.e., change from Fig.1a to Fig.1b. Therefore, now the loss is $-\Phi^+(G,C)$, which is shown in Fig.A for a good generator $G_{good}$ and different mode-collapsed generators $G_{collapsed}$ (designs of $G_{good}$ and $G_{collapsed}$ are the same as in our paper Sec.4). Comparing Fig.A and Fig.3a in our paper, the improvement using our proposed model can be observed: $G_{good}$ has

24 the lowest loss under our proposed model.

25 • **3. A separate discriminator to distinguish**
26 **real/fake rotated images:** We train this new model
27 **MS-v2**. Our preliminary result shows that this is very
28 competitive (Table A, row 1). We thank Reviewer for
29 this good suggestion. We will further analyze. We
30 note that this new model further corroborates our idea
31 to leverage discrimination of *rotated* real/fake images
32 to improve *generator* learning, as we have provided
33 theoretical and empirical evidence in our paper.

| Datasets | SS | MS | MS-v2 |
|---|---|---|---|
| **CIFAR-10** (10K-10K FID) | 12.37 | 11.40 | 11.15 |
| **CIFAR-100** (10K-5K FID) | 49.40 | 21.39 | - |
| **ImageNet 32×32** (10K-10K FID) | 26.04 (best) | 13.70 | - |
| **Stacked MNIST** (#mode) | $878.5 \pm 38.9$ | $943.2 \pm 31.4$ | - |
| **Stacked MNIST** (KL) | $0.99 \pm 0.19$ | $0.70 \pm 0.15$ | - |

Table A: Additional results. Baseline: Dist-GAN with ResNet, as in our paper. We also follow exactly the same experiment setup. **SS**: proposed in [4]; **MS**: this work. Note that for **mode coverage** experiment (row 4), our method **MS** achieves the best results for this dataset with tiny $K/4$ architecture, **outperforming state-of-the-art by a significant margin:** [36]: #mode = $859.5 \pm 68.7$, KL = $1.04 \pm 0.29$; [R1]: #mode = $859.5 \pm 36.2$, KL = $1.05 \pm 0.09$. We will update the paper with these results.

34 • 4. We show results of CIFAR-100 in Table A, row
35 2. We follow the setup in [R2] exactly, i.e. 10K-5K
36 FID. Note that [R2] has the state-of-the-art results for
37 CIFAR-100. As shown in Table A, row 2, **SS** suffers mode collapse (high FID), probably due to some classes like
38 "sunflowers", "fruit and vegetables", which original SS task does not encourage a generator to learn to produce them
39 (*rotated sunflowers are not rare*). Our **MS** achieves good FID on this dataset and outperforms [R2] (best 10K-5K FID =
40 23.6) on same dataset. Imagenet (128×128 or higher resolution) experiments require very extensive computation. So
41 far, most of Imagenet results are from big companies/institutions, eg., Google [2, 4, 41]. We will look for computing
42 resource for this experiment. We conduct the preliminary experiments on lower resolution Imagenet (32×32) as shown
43 in Table A, row 3. Our **MS** substantially outperforms **SS**. Note that training using our **MS** is also stable: **SS** suffers
44 from mode collapse (FID = 56.20) at the end of training and our **MS** attains the best FID at the end of training.

45 **Reviewer 3** • 1. We will further clarify Fig.3. Note some explanation is in L.167-180. • 2. These methods are compared
46 under **exactly the same settings** of [26, 4]. Some results are missing because previous methods did not report for these
47 experiments. • **3. No. of modes covered**: We thank Reviewer's suggestion. We follow exactly the same experiment
48 setup, tiny architecture $K/4$ and evaluation protocol of [25]. Table A, row 4 shows that our performance for mode
49 covered and KL divergence [25] is superior, significantly outperforms state-of-the-art [36, R1].

50 [R1] Karras et al., Progressive Growing of GANs for Improved Quality, Stability, and Variation, ICLR 2018.
51 [R2] Yamaguchi et al., Distributional Concavity Regularization for GANs, ICLR 2019.


[Meta-Review · NeurIPS 2019]

The paper addresses a problem in self supervised GAN, where the classes strictly have disjoint support. This is mitigated by introducing a new class for generated samples. Experiments are convincing.